# M^6^A Demethylase Inhibits Osteogenesis of Dental Follicle Stem Cells via Regulating miR-7974/FKBP15 Pathway

**DOI:** 10.3390/ijms242216121

**Published:** 2023-11-09

**Authors:** Linwei Zheng, Zhizheng Li, Bing Wang, Rui Sun, Yuqi Sun, Jiangang Ren, Jihong Zhao

**Affiliations:** 1State Key Laboratory of Oral & Maxillofacial Reconstruction and Regeneration, Key Laboratory of Oral Biomedicine Ministry of Education, Hubei Key Laboratory of Stomatology, School & Hospital of Stomatology, Wuhan University, Wuhan 430072, China; 2016302220031@whu.edu.cn (L.Z.); lzz2016lzz@whu.edu.cn (Z.L.); wangbing@whu.edu.cn (B.W.); sunrui12@whu.edu.cn (R.S.); sunyuqi@whu.edu.cn (Y.S.); 2Department of Oral and Maxillofacial Surgery, School & Hospital of Stomatology, Wuhan University, Wuhan 430072, China

**Keywords:** dental follicle stem cell, N6-methyladenosine, microRNA, FK506-binding protein, osteogenesis, actin cytoskeleton

## Abstract

N6-methyladenosine (m^6^A) is the most abundant RNA modification, regulating gene expression in physiological processes. However, its effect on the osteogenic differentiation of dental follicle stem cells (DFSCs) remains unknown. Here, m^6^A demethylases, the fat mass and obesity-associated protein (FTO), and alkB homolog 5 (ALKBH5) were overexpressed in DFSCs, followed by osteogenesis assay and transcriptome sequencing to explore potential mechanisms. The overexpression of FTO or ALKBH5 inhibited the osteogenesis of DFSCs, evidenced by the fact that RUNX2 independently decreased calcium deposition and by the downregulation of the osteogenic genes OCN and OPN. MiRNA profiling revealed that miR-7974 was the top differentially regulated gene, and the overexpression of m^6^A demethylases significantly accelerated miR-7974 degradation in DFSCs. The miR-7974 inhibitor decreased the osteogenesis of DFSCs, and its mimic attenuated the inhibitory effects of FTO overexpression. Bioinformatic prediction and RNA sequencing analysis suggested that FK506-binding protein 15 (FKBP15) was the most likely target downstream of miR-7974. The overexpression of FKBP15 significantly inhibited the osteogenesis of DFSCs via the restriction of actin cytoskeleton organization. This study provided a data resource of differentially expressed miRNA and mRNA after the overexpression of m^6^A demethylases in DFSCs. We unmasked the RUNX2-independent effects of m^6^A demethylase, miR-7974, and FKBP15 on the osteogenesis of DFSCs. Moreover, the FTO/miR-7974/FKBP15 axis and its effects on actin cytoskeleton organization were identified in DFSCs.

## 1. Introduction

Dental follicle stem cells (DFSCs) are multipotent stem cells capable of differentiation into osteoblasts, adipocytes, fibroblasts, and cementoblasts. The inhibition of osteogenesis in DFSCs results in abnormal tooth development and eruption [1,2]. The further elucidation of the mechanism of DFSC osteogenic differentiation is of great significance for the treatment of dental developmental diseases and the advancement of bone tissue engineering.

N6-methyladenosine (m^6^A) is the most abundant modification on polyadenylated RNA [3], and is the first-ever confirmed reversible RNA internal modification. M^6^A modification is dynamically regulated by m^6^A methylases such as methyltransferase-like 3 (METTL3), methyltransferase-like 14 (METTL14), and Wilms tumor 1-associating protein (WTAP), as well as m^6^A demethylases including fat mass and obesity-associated protein (FTO) and alkB homolog 5 (ALKBH5) [4,5,6,7]. FTO was the first identified demethylase that could remove the m^6^A modification in the internal (m^6^A) and 5′cap (m^6^A_m_) of mRNAs [8]. FTO regulates the expression of osteogenic genes such as RUNX2 and Osterix by affecting mRNA translation and decay or pre-mRNA splicing and polyadenylation to directly or indirectly modulate osteogenic differentiation in diverse pluripotent stem cell lines [9,10,11,12]. Furthermore, FTO demethylase influences the fate of modified RNA molecules, not only in mRNAs, but also in non-coding RNAs [13]. Abundant evidence has demonstrated that miRNAs are involved in bone homeostasis [14]. However, the role of miRNAs in FTO demethylase-regulated osteogenic differentiation remains largely unclear.

Our present study was designed to investigate the roles of m^6^A demethylases in DFSCs. We found that the upregulation of FTO and ALKBH5 inhibited the osteogenic differentiation of DFSCs. Mechanistically, the overexpression of FTO and ALKBH5 accelerated miR-7974 degradation in DFSCs, which was followed by the upregulation of FKBP15 and the inhibition of actin filament remodeling.

## 2. Results

### 2.1. Overexpression of FTO and ALKBH5 Inhibited the Osteogenic Capacity of DFSCs

To investigate the roles of demethylases in DFSCs, we overexpressed FTO and ALKBH5 in DFSCs to explore their effects on osteogenic differentiation. The mRNA and protein levels of FTO or ALKBH5 were elevated in FTO-overexpressing DFSCs (FTO-OE) or ALKBH5-overexpressing DFSCs (ALKBH5-OE), respectively, confirming their overexpression in DFSCs (Figure 1A,B). After being induced in osteogenic differentiation medium for 12 days, alizarin red staining showed that the calcium deposits that were produced by FTO-OE or ALKBH5-OE DFSCs were significantly decreased compared to the control DFSCs (Figure 1C). The expression levels of osteogenesis-related genes, osteocalcin (OCN), and osteopontin (OPN) were consistently downregulated in FTO-OE or ALKBH5-OE DFSCs compared with the control DFSCs (Figure 1D). Conversely, in METTL14-overexpressing DFSCs (METTL14-OE), the calcium deposits and OCN and OPN expressions were also upregulated compared to the control DFSCs (Figure 1E). However, Runt-related transcription factor 2 (RUNX2), an important osteogenic gene, was not significantly changed either in FTO-OE or ALKBH5-OE DFSCs (Appendix A), indicating that the effects of FTO or ALKBH5 on the osteogenic differentiation of DFSCs were RUNX2-independent.

### 2.2. Downregulation of miR-7974 Was Involved in Decreased Osteogenic Capacity of FTO-OE or ALKBH5-OE

MiRNA sequencing was performed on FTO-OE or ALKBH5-OE DFSCs (Figure 2A,B) to explore novel patterns of m^6^A demethylase regulating osteogenic differentiation. In the intersection of differentially expressed miRNAs, we identified 10 miRNAs that shared a common alteration tendency in both FTO-OE and ALKBH5-OE DFSCs (Figure 2C). Among these miRNAs, most of them were downregulated, with miR-7974 being the most significantly downregulated compared to the control DFSCs. Further validation of miR-7974 stability using ActD showed that miR-7974 stability was significantly decreased in FTO-OE or ALKBH5-OE DFSCs (Figure 2D). The expression of miR-7974 by RT-PCR was consistent with that detected using high-throughput technology (Figure 2E).

To demonstrate the role of miR-7974 in m^6^A demethylase mediated osteogenic inhibition, miR-7974 mimic was transfected into FTO-OE DFSCs. After 12 days of osteogenesis induction, alizarin red staining showed that miR-7974 mimic significantly attenuated the inhibitory effects of FTO overexpression on calcium deposits in DFSCs (Figure 2F). The expression levels of OCN and OPN were partially restored in FTO-OE DFSCs after the latter were transfected with miR-7974 mimic (Figure 2G). Further validation using the miR-7974 inhibitor revealed a decrease in calcium deposition and downregulation of OCN and OPN in DFSCs (Figure 2H,I). These data indicated that the inhibitory effects of m^6^A demethylase on osteogenesis in DFSCs were at least partially miR-7974-dependent. 

### 2.3. FKBP15 Is Increased in FTO-Overexpressing DFSCs and Targeted by miR-7974

To elucidate the downstream of miR-7974 that negatively regulated osteogenic differentiation in DFSCs, we predicted miR-7974 target genes using miRNet and TargetScan databases. By means of overlap-screening the prediction results with those significantly downregulated genes in both FTO-OE cells and ALKBH5-OE cells acquired by performing transcriptome profiling (Appendix A), the three most downregulated genes (FKBP15, SERF1A, and KIAA0040) were identified at the intersection (Figure 3A). Further validations were performed using the TargetScan database for these three genes, and FK506-binding protein 15 (FKBP15, also known as WAFL, FKBP133, or KIAA0674) was identified as the most likely direct target downstream of miR-7974 (Figure 3B). RT-PCR and Western blot data showed that FKBP15 was significantly decreased in DFSCs transfected with miR-7974 mimics, while the miR-7974 inhibitor upregulated the FKBP15 level (Figure 3C). More importantly, in FTO-OE DFSCs, FKBP15 was significantly upregulated compared to the control DFSCs, and miR-7974 mimics attenuated the effects of FTO overexpression on FKBP15 expression (Figure 3D).

### 2.4. FKBP15 Inhibits Osteogenic Capacity of DFSCs by Interacting with Actin Cytoskeleton

Next, we silenced FKBP15 using siRNA to determine the role of FKBP15 in the osteogenic differentiation of DFSCs. The efficiency of FKBP15-specific siRNA was confirmed by performing Western blot analysis and RT-PCR on both the control and the FTO-OE DFSCs (Figure 4A). After being induced in osteogenesis differentiation medium and stained with alizarin red, the DFSCs transfected with FKBP15-specific siRNA formed more calcium deposits and expressed higher levels of OCN and OPN (Figure 4B,C). Moreover, the inhibitory effects of FTO overexpression on osteogenic differentiation in DFSCs were dramatically attenuated after the knockdown of FKBP15 (Figure 4D–F). Actin cytoskeleton organization plays a critical role during osteogenesis [15]. Considering previous studies about the function of FKBP15 in actin cytoskeleton remodeling [16], we further performed phalloidin staining on DFSCs. Our immunofluorescence data showed a dotted and stronger cytoplasmic staining of FKBP15 in the FKBP15-OE DFSCs compared to the control DFSCs (Figure 5A). During osteogenic differentiation, the control DFSCs transformed from a fibroblast-like phenotype to a star-like shape with numerous spreading filopodia. By sharp contrast, the FKBP15-OE DFSCs became square-like in shape with a lack of spreading filopodia (Figure 5A,B). Moreover, the circularity was significantly higher in the control DFSCs than in the FKBP15-OE DFSCs (Figure 5C). During osteogenic differentiation, the actin cytoskeleton changed from many thin and parallel microfilament bundles extending across the entire cytoplasm to a few thick filament bundles around the periphery of differentiated cells in the control DFSCs (Figure 5D). However, no significant remodeling of the actin filament was observed in the FKBP15-OE cells (Figure 5D). Alizarin red staining and RT-PCR consistently confirmed that calcium deposits and the expression of OCN and OPN were significantly decreased in the FKBP15-OE DFSCs compared to the control DFSCs (Figure 5E,F).

## 3. Discussion

DFSCs represent a rich source population of pluripotent stem cells derived from dental follicles, playing a crucial role in tooth development and eruption [17,18]. DFSCs exhibit a strong osteogenic capacity, making them an expected seed cell for bone regeneration and tissue engineering [19,20]. Though the regulatory mechanism of osteogenic differentiation in DFSCs shares many similarities with that of bone marrow stem cells (BMSCs), some differences do exist. 

Cytoskeleton remodeling has been widely proven to be integral to the commitment of pre-osteoblasts such as BMSCs [15,21,22]. Alterations in cytoskeletal integrity and cell shape dictate the expressions of osteogenic genes, the deposition of extracellular matrix, and calcification and were essential during osteogenic differentiation [22,23,24,25]. Cytoskeleton remodeling itself is believed to exert its effects on osteogenic differentiation, independently from soluble differentiation factors [15]. However, the regulation of osteogenesis derived from cytoskeletal system in DFSCs remains largely unexplored. In the present study, we demonstrated that obstructing actin filament system remodeling and cell shape changes inhibited the osteogenic differentiation of DFSCs, which provides evidence for the involvement of the microfilaments of the cytoskeleton in the osteogenesis of DFSCs.

FK506-binding protein was initially identified as one of two major immunophilins that bind the immunosuppressive drug FK506 and exhibit peptidylprolyl cis/trans isomerase activity. Small size FKBPs contain only an FK506-binding domain, while FKBPs with large molecular weights possess extra domains [26]. FKBP15, also known as FKBP133, KIAA0674 [27], or WAFL [16], was characterized as containing Wiskott–Aldrich syndrome protein homology region 1 (WH1) domain and FK506-binding domain in 2006 [27]. FKBP15 interacted with the actin filament system instead of the microtubules system through the WH1 domain, participating in the transport of endosomal cargo [16]. In mouse dorsal root ganglion neurons, FKBP15 co-localized with F-actin. The overexpression of mutant FKBP15, where the WH1 domain is deleted, results in a reduction in the growth cone size and the number of filopodia, restricting cell reshaping and spreading [27]. FKBP15 affects both intracellular cargo transport and changes in cell shape and spread, both of which are important for osteogenic differentiation. Immunofluorescence showed that FKBP15 was widely co-localized with actin filaments in DFSCs. The overexpression of FKBP15 inhibited the reorganization of the actin filament system, cell reshaping, and spreading, accompanied by a downregulation of osteogenic-related genes (OCN and OPN) and a decrease in calcium deposits in DFSCs. This study demonstrated that the overexpression of FKBP15 inhibited the osteogenesis of DFSCs, as the constraint of actin cytoskeleton remodeling affected osteogenic differentiation. 

Research on miRNA regulation of osteogenic differentiation has been widely reported [28,29], including within the context of DFSCs [30,31]. In most studies, RUNX2 acted as a central factor during the osteogenesis of DFSCs, either influencing miRNA or being influencing by it [30,31]. In parallel, emerging studies have elucidated the existence of RUNX2-independent pathways that govern the osteogenesis of DFSCs [32,33,34]. For instance, dexamethasone alone induced mineralization and OCN expression in DFSCs via the ZBTB16 (zinc finger and BTB domain containing protein 16)/NR4A3 (nuclear receptor subfamily 4 group A member 3) axis but did not stimulate RUNX2 [35]. Yet in BMSCs, the expression of ZBTB16 was essential for inciting RUNX2 and fostering osteogenic differentiation [36]. These suggested that DFSCs and BMSCs harbored different mechanistic bases for osteogenic differentiation [37]. In this study, we presented a novel pattern of DFSC osteogenic differentiation regulation that operated independently of RUNX2. To our knowledge, neither the role of miR-7974 in bone metabolism nor its targeting of FKBP15 has been previously reported. Here, miR-7974 emerged as the top differentially regulated gene following m^6^A demethylase overexpression. The overexpression of m^6^A demethylases significantly reduced the stability of miR-7974, possibly leading to a decrease in the miR-7974 level. Whether the accelerated degradation of miR-7974 is directly related to its own m^6^A level remains to be explored. The miR-7974 inhibitor directly increased FKBP15 expression and inhibited osteogenic differentiation and the calcium deposition of DFSCs, while an miR-7974 mimic produced the opposite effects. Thus, miR-7974 plays an important role in the osteogenic differentiation of DFSCs, independent of RUNX2. Although the poor conservation of miR-7974 among different species limited the feasibility of animal experiments in this study, its effects on FKBP15 and cytoskeletons open a new avenue for exploring stem cell commitment.

N6-methyladenosine is one of the most abundant internal mRNA modifications; its alteration exerts a more complex influence on diverse cellular bioprocesses [38,39]. The effects of m^6^A on RNA are mediated by m^6^A readers (methylases, such as METTL3 and METTL14) and m^6^A erasers (demethylases, such as ALKBH5 and FTO), as well as m^6^A writer-complex components [39]. The role of m^6^A in osteogenic differentiation is currently under extensive research, but its function varies among different cell lines. In human BMSCs, a high level of m^6^A was found to promote osteogenic differentiation by upregulating RUNX2 and Osterix [10], while in mouse BMSCs and human MSCs derived from osteoporosis patients, the knockout of demethylase FTO decreased osteogenic differentiation [40]. The effects of m6A on osteogenic differentiation in DFSCs have not been studied. In this study, the overexpression of demethylases FTO and ALKBH5 significantly reduced the calcium deposition and the expressions of OCN and OPN in DFSCs. By contrast, the overexpression of methylase METTL14 enhanced these events. However, there was no significant change in RUNX2 expression level in either demethylases or methylase-overexpressing DFSCs. Therefore, we conducted miRNA and mRNA sequencing on two different overexpressing cell phenotypes, revealing an miR-7974/FKBP15 pathway regulating DFSC osteogenesis that was independent from RUNX2. Though RUNX2 and Osterix are both key transcription factors in the process of osteogenic differentiation in DFSCs, neither of them is involved in the regulation of FTO-related osteogenesis regulation in DFSCs. Due to cellular heterogeneity, additional key transcription factors probably act in DFSC osteogenesis, which requires further analysis of our transcriptome results. On the other hand, the biophysical regulations of filament cytoskeletons and cell morphology possibly act as a separate regulation independent of molecular level mechanisms. To our knowledge, no research has been conducted on the effects of m^6^A on cytoskeletons. Here, we preliminarily demonstrated the effects of m^6^A modifications on cytoskeletal remodeling. The overexpression of m^6^A demethylase FTO inhibited the recombination of the actin filament system via an miR-7974/FKBP15 pathway, affecting intracellular and extracellular cargo transport reliant on the actin filaments, restricting cell reshaping, and limiting filopodia formation during osteogenic differentiation in DFSCs.

This study has limitations. This study primarily investigated the effects of m^6^A demethylase FTO on the osteogenesis of DFSCs as an example, while merely performing phenomenal observations of the effects of other m^6^A demethylases, ALKBH5 and m^6^A methylase METTL14, on osteogenic capacity. Consequently, the miR-7974/FKBP15 pathway may not fully illustrate the situation when the overall intracellular m^6^A modification level changes. Global m^6^A methylation levels, and the expression of methylases and demethylases in the process of osteogenic differentiation, remain to be investigated. In addition, the stability of miR-7974 was decreased after the overexpression of FTO and ALKBH5 evidenced by ActD assay, and further experiments are required to understand whether m^6^A demethylases directly affect its methylation modification, given that demethylase FTO also participates in m^6^A_m_ [39,41] and m^1^A [8] in non-coding RNAs. The specific mechanisms of FKBP15 binding to the actin cytoskeleton and affecting its recombination need additional validation in future animal experiments.

In summary, this study elucidated a mechanism by which m^6^A modification regulated the osteogenic differentiation of DFSCs. The overexpression of m^6^A demethylase FTO reduced the stability of miR-7974 and decreased its level, which increased the expression of FKBP15 and impacted the reorganization of the actin filament system and inhibited the osteogenic differentiation of DFSCs. This could potentially offer therapeutic strategies for dental developmental diseases and provide an experimental basis for bone tissue engineering.

## 4. Materials and Methods

### 4.1. Cell Culture

DFSCs isolated from fresh dental follicle tissues of human third molars were donated by the Oral Stem Cell Bank (Beijing, China) with the signed and fully informed consent of the donors. The research protocol was approved by the review board of the Ethics Committee of the Hospital of Stomatology, Wuhan University (Protocol Approval 2022A36), and was conducted in accordance with the principles of the Declaration of Helsinki. All participants signed an informed consent document before tissue collection. DFSCs were cultured in an α-MEM medium consisting of 10% fetal bovine serum (Gibco, Erie County, NY, USA) and 1% penicillin–streptomycin solution (Sigma-Aldrich LLC., Saint Louis, MO, USA) and were incubated in a humidified atmosphere containing 5% CO_2_ at 37 °C. The cell culture medium was refreshed every 2 days, and primary cells from passages 3–5 were used for experiments as previously described [2].

### 4.2. Cell Transfection

The recombinant overexpression plasmids related to this study were constructed by Syngentech (Beijing, China). The DFSCs were transfected with the lentivirus LV-hef1a-mNeongreen-P2A-Puro-WPRE-CMV-FTO (human, NM_001080432)-3xflag, pLV-hef1a-mNeongreen-P2A-Puro-WPRE-CMV-ALKBH5 (human, NM_017758)-3xflag, and pLV-hef1a-mNeongreen-P2A-Puro-WPRE-CMV-METTL14 (human, NM_020961)-3xflag, after which selection was performed using 2.5 μg/mL puromycin for 6 days. Synthetic miR-7974 mimics and inhibitors, and siRNA oligonucleotides for FKBP15, were synthesized and transfected using a RiboFECT CP Transfection Kit (RiboBio Co., Ltd., Guangzhou, China). The efficiency of overexpression or downregulation was examined by performing RT-PCR and Western blot analysis. 

### 4.3. Osteogenesis Induction

As previously [2], the DFSCs were cultured for 12 days in the osteogenesis differentiation medium containing α-MEM, 10% FBS, 5 μg/mL insulin, 0.1 μM dexamethasone, 0.2 mM L-ascorbate-2 phosphate, and 10 mM β-glycerophosphate. The medium was refreshed every 2 days. After 12 days, the cells were fixed with 4% paraformaldehyde and stained with alizarin red (Sigma-Aldrich, Darmstadt, GER). The mineral depositions were photographed using light microscopy (Leica, HDB, GER). The staining was analyzed by two independent observers using the Image Pro Plus version 6.0 software (Media Cybernetics). All the analyses were carried out in a blinded manner. For the osteogenic induction of siFKBP15-DFSC, siRNA was added again on the 7th day to maintain the silence of the FKBP15 expression.

### 4.4. High Throughput Sequencing of mRNA and microRNA

After the transfected efficiency examination, FTO-OE or ALKBH5-OE DFSCs were seeded in 6 cm dish and cultured to 80% confluency in an α-MEM medium consisting of 10% fetal bovine serum and 1% penicillin–streptomycin solution; then, the total RNA was collected. RNA degradation and contamination was monitored on 1% agarose gels. RNA purity was checked using the NanoPhotometer^®^ spectrophotometer (Implen U.S.A., Inc., Westlake Village, CA, USA). RNA concentration was measured using a Qubit^®^ RNA Assay Kit in Qubit^®^ 2.0 Flurometer (Life Technologies Corp., Mountain View, CA, USA). RNA integrity was assessed using the RNA Nano 6000 Assay Kit of the Bioanalyzer 2100 system (Agilent Technologies Inc., Folsom, CA, USA).

A total amount of 3 μg RNA per sample was used as input material for the RNA sample preparations. Firstly, ribosomal RNA was removed using an Epicentre Ribo-zero™ rRNA Removal Kit (Epicentre Technologies Corp., Madison, WI, USA), and rRNA free samples were purified by performing an ethanol precipitation. Subsequently, sequencing libraries were generated using the rRNA depleted RNA by employing the NEBNext^®^ Ultra™ Directional RNA Library Prep Kit for Illumina^®^ (New England Biolabs, Ipswich, MA, USA), following manufacturer’s recommendations. Briefly, fragmentation was carried out using divalent cations under an elevated temperature in NEBNext First Strand Synthesis Reaction Buffer (5X). First strand cDNA was synthesized using a random hexamer primer and M-MuLV Reverse Transcriptase (RNaseH-). Second strand cDNA synthesis was subsequently performed using DNA Polymerase I and RNase H. In the reaction buffer, dNTPs with dTTP were replaced by dUTP. Remaining overhangs were converted into blunt ends via exonuclease/polymerase activities. After adenylation of the 3′ ends of DNA fragments, NEBNext Adaptors with hairpin loop structure were ligated to prepare for hybridization. In order to select cDNA fragments of preferentially 150–200 bp in length, the library fragments were purified with the AMPure XP system (Beckman Coulter, Beverly, MA, USA). Then, 3 μL USER Enzyme (NEB, USA) was used with size-selected, adaptor-ligated cDNA at 37 °C for 15 min followed by 5 min at 95 °C before PCR. Then, PCR was performed using Phusion High-Fidelity DNA polymerase, Universal PCR primers, and Index (X) Primer. Finally, the products were purified (AMPure XP system), and their library quality was assessed using the Agilent Bioanalyzer 2100 system. The clustering of the index-coded samples was performed on a cBot Cluster Generation System using TruSeq PE Cluster Kit v3-cBot-HS (Illumia), according to the manufacturer’s instructions. After cluster generation, the libraries were sequenced on an Illumina Noveseq platform, and 150 bp paired-end reads were generated.

### 4.5. Real-Time Quantitative PCR

The total RNA was extracted and reverse transcribed into cDNA using the E.Z.N.A. Total RNA Kit (Omega Bio-tek, Norcross, GA, USA) and the First-Strand cDNA Synthesis kit (Thermo Fisher, Norristown, PA, USA). Quantitative real-time PCR (RT-PCR) was performed on the Quant Studio 6 Flex Real-Time PCR System (Applied Biosystems, South San Francisco, CA, USA) using the Fast Start Universal SYBR Green Master (Roche Molecular Systems, Branchburg, NJ, USA). The primer sequences are shown in Table 1. The RT-PCR primers specific to miR-7974 were designed by RiboBio (RiboBio Inc., Guangzhou, China).

### 4.6. MiRNA Stability Assay

FTO-OE or ALKBH5-OE cells were seeded in 24-well plates (5  ×  10^4^ cells per well). Twenty-four hours later, Actinomycin D (ActD, 5 μg/mL) (Sigma-Aldrich LLC., Saint Louis, MO, USA) was added to general culture medium, followed by incubation for 2 h, 4 h, 6 h, or 8 h. The total RNA was extracted, and the miR-7974 level was detected by performing RT-PCR. 

### 4.7. Western Blot Analysis

Equal amounts of total proteins (20 μg) were separated by 10% or 12% SDS-PAGE and transferred onto PVDF membranes, which were blocked with 5% nonfat milk, probed with primary antibodies for FTO (ET1705-89, Huabio, Hangzhou, China), ALKBH5 (ab195377, Abcam plc, Cambridge, UK), FKBP15 (ab14432, Abcam plc, Cambridge, UK), and GAPDH (ab8245, Abcam plc, Cambridge, UK) and then incubated with horseradish peroxidase-conjugated secondary antibodies. Immunoreactive bands were visualized using a chemiluminescence substrate (Pierce, Greenfield, MA, USA). The band intensity was calculated using Image J software v1.8.0 (National Institutes of Health, Bethesda, MD, USA).

### 4.8. Immunofluorescence

After being blocked with 2% bovine serum for 2 h, FKBP15 (1:200) and phalloidin (1:200, ab176757, Abcam plc, Cambridge, UK) antibodies were diluted and added to the slices. After being incubated with mixed secondary antibodies, all of the slices were covered with sealing agent containing DAPI (P0131, Beyotime, Shanghai, China). Fluorescence images were captured using a confocal microscope (TCS SP8, Leica, Nussloch, Germany). The fluorescent intensities were measured using Image J software.

### 4.9. MiRNA Target Prediction

Candidate miRNAs that were selected for RNA-seq analysis were utilized for target prediction using the publicly available TargetScan (http://www.targetscan.org, accessed on 12 December 2022) and miRnet (https://www.mirnet.ca, accessed on 12 December 2022). The TargetScan database was used to predict the potential binding sites for miRNA-7974 in the FKBP15 mRNA 3′-UTR. The genes that were predicted as candidate miRNA targets and those that were selected on the basis of VENNY 2.1.0 (http://liuxiaoyuyuan.cn, accessed on 12 December 2022) were compared, and the genes that appeared in the two lists were included in the present study.

### 4.10. Data and Statistical Analysis

Data were presented as mean ± SD and analyzed using GraphPad Prism 10.0 (GraphPad Software, San Diego, CA, USA). Statistical significance was calculated by using Student’s *t*-test when comparing two groups and one-way analysis of variance followed by post hoc tests when comparing more than two groups. A level of *p* < 0.01 was considered statistically significant.

## Figures and Tables

**Figure 1 ijms-24-16121-f001:**
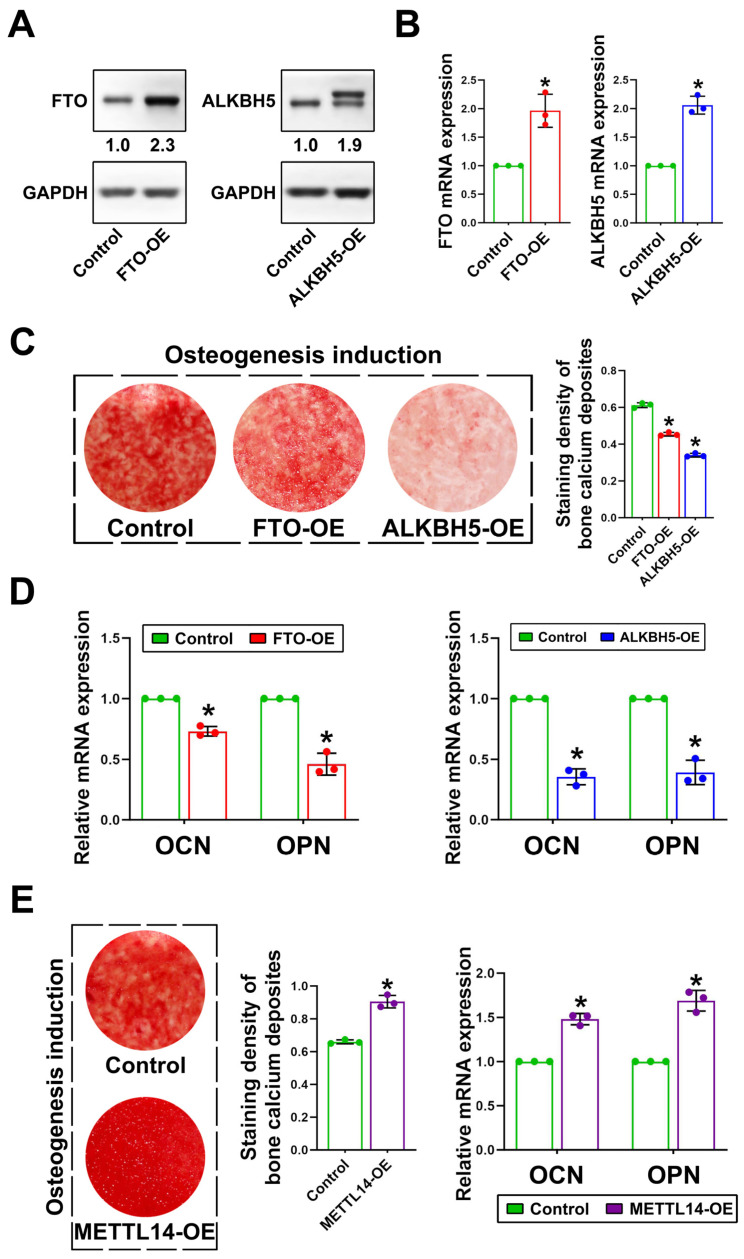
Osteogenic capabilities of DFSC were decreased after overexpression of FTO and ALKBH5. (**A**,**B**) The efficiency of FTO and ALKBH5 overexpression was validated by performing Western blot analysis and RT-PCR. (**C**) Alizarin red staining showed decreased calcium deposit formation in FTO and ALKBH5-overexpressing DFSCs. (**D**) Expressions of osteogenesis related genes OCN and OPN were downregulated in FTO-OE and ALKBH5-OE DFSCs. (**E**) Overexpression of METTL14 increased its calcium deposition and OCN and OPN expressions in DFSCs. * *p* < 0.01, n = 3.

**Figure 2 ijms-24-16121-f002:**
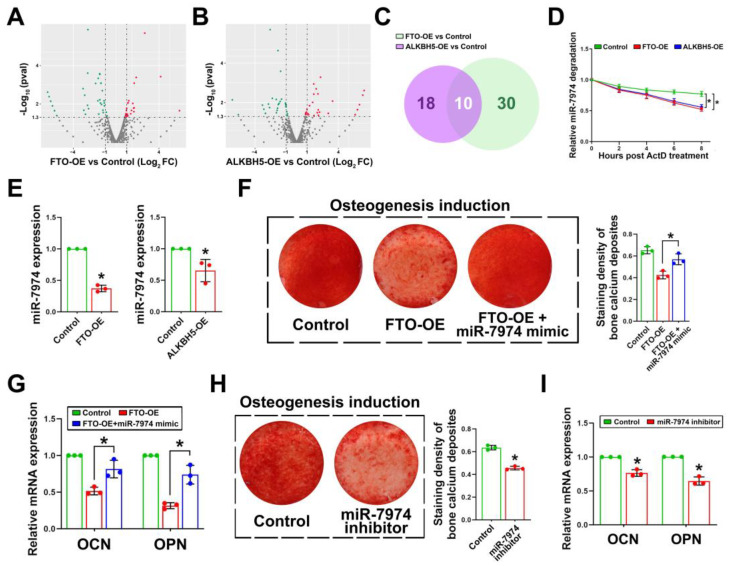
Analysis of miRNA profiling in FTO-OE and ALKBH5-OE DFSCs and the effects of miR-7974 on the osteogenic differentiation of DFSCs. (**A**) Volcano plot of the Log_2_ (fold change) between FTO-OE and vehicle control and the negative Log_10_ (*p*-value). (**B**) Volcano plot of the Log_2_ (fold change) between ALKBH5-OE and vehicle control and the negative Log_10_ (*p*-value). (**C**) 10 miRNAs were identified in the intersection of differentially expressed miRNAs between FTO-OE and ALKBH5-OE DFSCs. (**D**) RT-PCR was conducted to show the stability of miR-7974 after adding ActD. (**E**) Level of miR-7974 was downregulated in both FTO-OE and ALKBH5-OE DFSCs. (**F**) miR-7974 mimic reversed the osteogenic capability of DFSCs inhibited by FTO overexpression. (**G**) Expressions of OCN and OPN were rescued after treatment with miR-7974 mimic in FTO-OE DFSCs. (**H**) The osteogenic capability of DFSCs was decreased after treatment with miR-7974 inhibitor. (**I**) Treatment with miR-7974 inhibitor downregulated OCN and OPN expression in DFSCs. * *p* < 0.01, n = 3.

**Figure 3 ijms-24-16121-f003:**
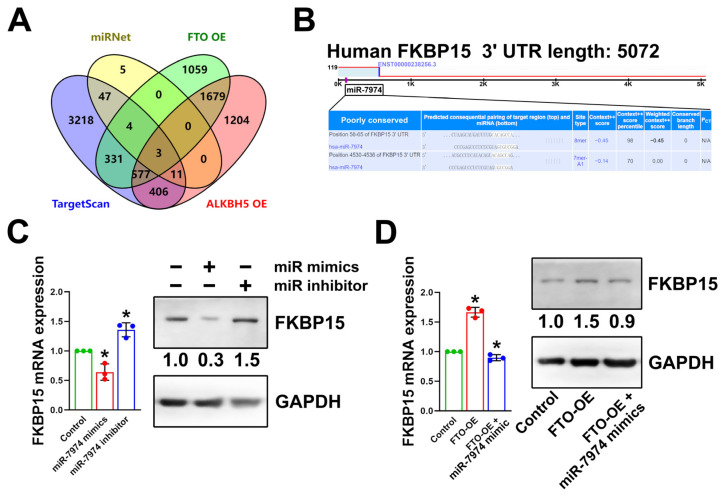
FKBP15 was the target of miR-7974, and was upregulated in FTO-OE DFSCs. (**A**) FKBP15 was identified as the most likely target gene at the intersection among lists of miR-7974 target predictions and differentially expressed genes in FTO-OE and ALKBH5-OE DFSCs. (**B**) Predicted binding sites of FKBP15 and miR-7974 identified using TargetScan. (**C**) Protein and mRNA levels of FKBP15 were strongly regulated by miR-7974 mimic and inhibitor. (**D**) Expression of FKBP15 was upregulated in FTO-OE DFSCs and could be decreased by miR-7974 mimic. * *p* < 0.01, n = 3.

**Figure 4 ijms-24-16121-f004:**
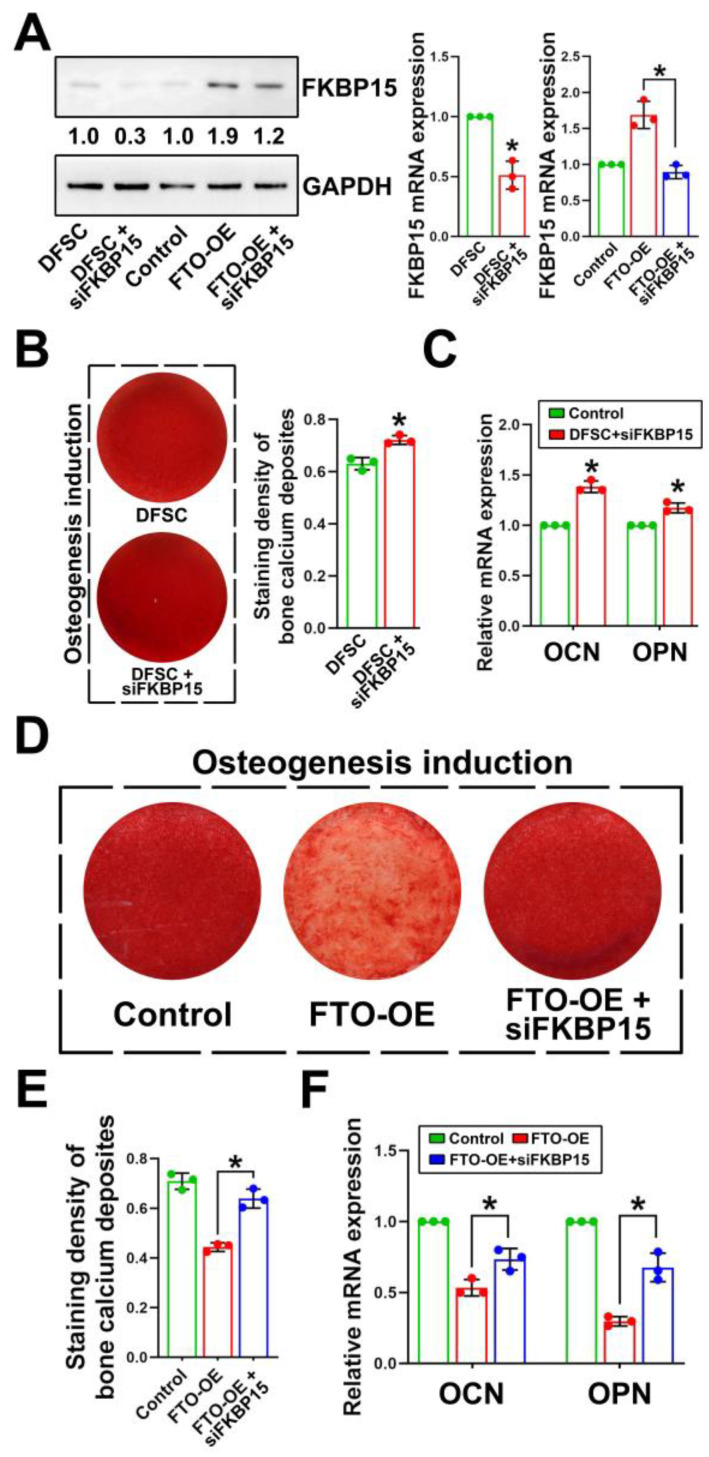
FKBP15 was upregulated and participated in the FTO-overexpression-induced inhibition of osteogenic differentiation in DFSCs. (**A**) Validation of the siRNA efficiency of FKBP15. (**B**,**C**) Silencing of FKBP15 slightly increased calcium deposition and OCN and OPN expression of DFSCs. Knockdown of FKBP15 restored osteogenic capacity of FTO-OE DFSCs evidenced by calcium deposit staining (**D**,**E**) and expressions of OCN and OPN (**F**). * *p* < 0.01, n = 3.

**Figure 5 ijms-24-16121-f005:**
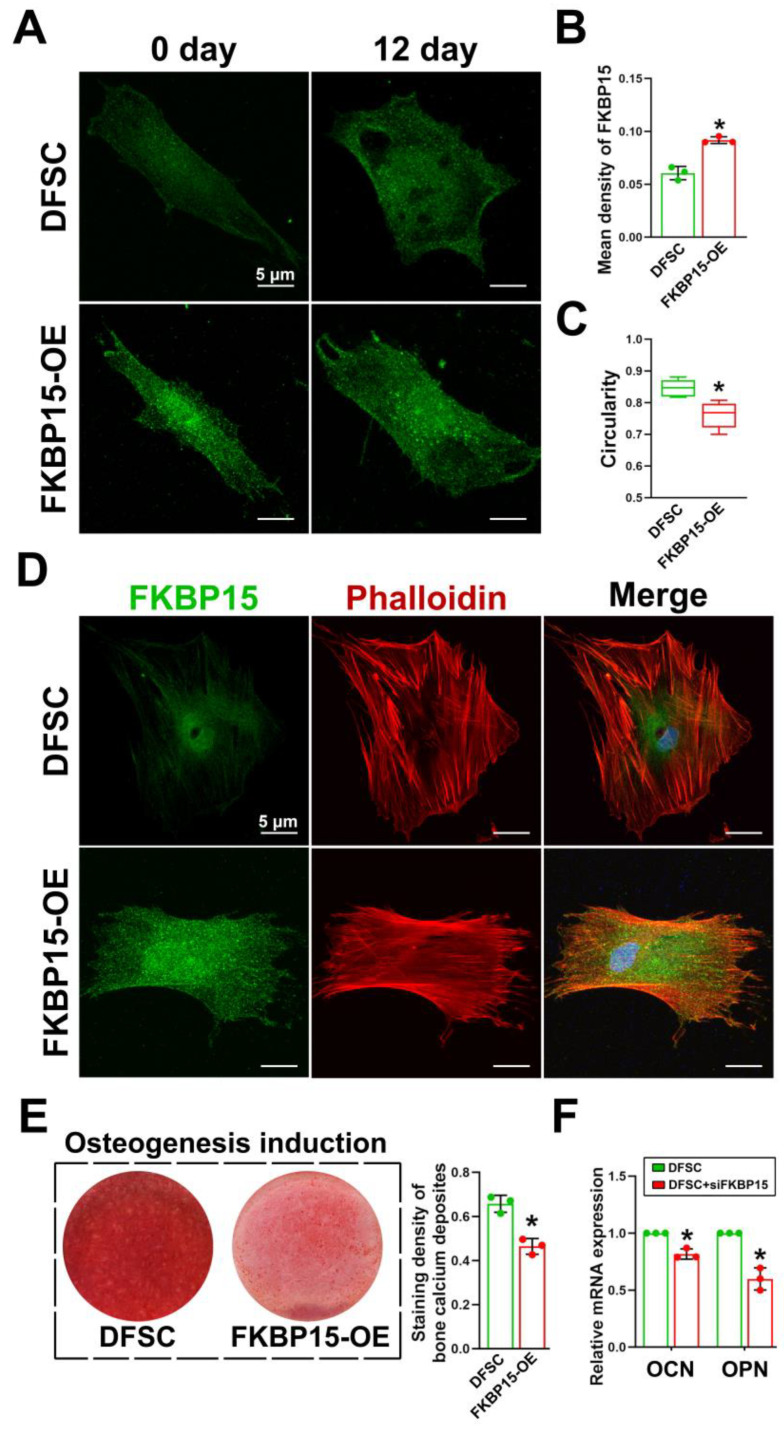
Overexpression of FKBP15 inhibited osteogenic differentiation of DFSCs by affecting actin cytoskeleton and cell reshaping during the induction. (**A**) FKBP15 has a dotted cytoplasmic staining in DFSCs. The shape of control DFSCs was changed from fibroblast-like phenotype to star-like shape with numerous spreading filopodia, while phenotype of FKBP15-OE DFSCs was not significantly changed or became square-like in shape and lacking in filopodia. (**B**) Semiquantitative analysis of fluorescence intensity of FKBP15 in DFSCs. (**C**) Cell circularity of control group was significantly higher than that of FKBP15-OE group. (**D**) Twelve days after induction, actin cytoskeleton changed from many thin and parallel microfilament bundles extending across the entire cytoplasm to a few thick filament bundles around the periphery in DFSC group, while in FKBP15-OE group, no significant remodeling of actin filament was observed. (**E**,**F**) Calcium deposits and expressions of OCN and OPN were significantly decreased in FKBP15-OE group compared to control DFSC group. * *p* < 0.01, n = 3.

**Table 1 ijms-24-16121-t001:** Primer sequences used for real-time PCR.

Gene	Forward (5′-3′)	Reverse (5′-3′)
FTO	AGCATGGCTGCTTATTTCGG	GAGCCTGGTGTTCAGGTACT
ALKBH5	AGATCGCCTGTCAGGAAACA	GACTTGCGCCAGTAGTTCTC
METTL14	AATCGCCTCCTCCCAAATCT	CCACCTCTTTCTCCTCGGAA
FKBP15	ACCACAGATGACCTCACAGG	AGCTGAGTGGGAATCGAGAG
OCN	ATGAGAGCCCTCACACTCCT	CTTGGACACAAAGGCTGCAC
OPN	ACACATATGATGGCCGAGGT	CTCGCTTTCCATGTGTGAGG
RUNX2	CCTCGGAGAGGTACCAGATG	GGTGAAACTCTTGCCTCGTC
Osterix	CATCTGCCTGGCTCCTTG	CAGGGGACTGGAGCCATA
GAPDH	TGCCCAGAACATCATCCCTG	GATACATTGGGGGTGGGGAC

## Data Availability

The data presented in this study are available on request from the corresponding author.

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
