# Peer review of "M6A Demethylase Inhibits Osteogenesis of Dental Follicle Stem Cells via Regulating miR-7974/FKBP15 Pathway"

_ijms, 2023, doi:10.3390/ijms242216121_

Round 1

Reviewer 1 Report

Comments and Suggestions for Authors

N6-methyladenosine (m6A) is an RNA modification and is involved in gene expression. It could therefore also be involved in biological processes such as osteogenic differentiation. However, the impact on osteogenic differentiation of DFSCs remains unknown. This study is a data source for differentially expressed miRNA and mRNA after overexpression of m6A demethylases in DFSCs. The authors demonstrated RUNX2-independent effects of m6A demethylase, miR-7974 and FKBP15 on osteogenesis of DFSCs. Furthermore, the FTO (or ALKBH5)/miR-7974/FKBP15 axis and its effects on the organization of the actin cytoskeleton in DFSCs were identified.

This study is very comprehensive and contributes new insights. Unfortunately, this working group does not adequately integrate its results with the research results of other working groups, which have also shown RUNX2-independent mechanisms during the osteogenic differentiation of DFSCs. The introduction and the discussion would benefit from this. There is also a lack of further investigations, such as what effects the miR-7974/FKBP15 axis has on the TF ZBTB16. A major revision would be necessary.

Reviewer 2 Report

Comments and Suggestions for Authors

Dear authors,

Please see the attached one.

Round 2

Reviewer 1 Report

Comments and Suggestions for Authors

no comment

Reviewer 2 Report

Comments and Suggestions for Authors

Dear authors,

Overall, there have been sufficient improvements and responses to the suggested comments. Therefore, I have no further comments or remarks on this manuscript.